# Full immunization coverage and associated factors among children aged 12–23 months in Somali Region, Eastern Ethiopia

Zemenu Shiferaw Yadita[1]*, Liyew Mekonen Ayehubizu[2]

1 Department of Reproductive Health and Population Studies, College of Medicine and Health Science, Bahirdar University, Bahirdar, Ethiopia, 2 Departments of Public Health, College of Medicine and Health Science, Jigjiga University, Jijiga, Ethiopia

* zaion2307@gmail.com

**Data Availability Statement:** All data supporting the findings is submitted with the manuscript. The data set for this article is Openly accessible without restriction (S2 File).

## Abstract

### Objective

Despite those efforts in expanded programs of immunization, nearly one fifth of children in developing countries miss out basic vaccines. Moreover, many children who started vaccination fail to complete immunization.Identifying associated factorswhich is scarce in the study area, is crucial for interventions. This study assessed full-immunization and associated factors among children aged 12–23 months in Somali region, Eastern Ethiopia.

### Methods

A community-based cross-sectional study design was conducted from October 1–30, 2018, in selected rural and urban kebeles in Somali regionamong 612 children. Cluster sampling was employed and data was collected using structured questionnaire. Full-immunization was measured by maternal recall and vaccination card.Data entry and analysis was done by EpiData3.1 and SPSSversion.20 respectively. Binary logistic regression with Bivariate and Multivariable model was usedto identify predictors of full-immunization. Odd ratios were computed and P-value <0.05 was considered as statistically significant.

### Results

Based on maternal recall plus vaccination card 249(41.4%) of children were completed immunization, while vaccination only by card was 87(29.7%). Only 238(39.5%) of participants had good knowledge about vaccination. Not knowing to come back for next visits 197(55.8%) were the major reason for dropout. Residing in urban (AOR = 2.0, 95%CI: 1.0, 3.9),primary educated mothers(AOR = 2.2, 95%CI: 1.0, 5.0), married mothers (AOR = 4.2, 95%CI:1.0, 18), higher average monthly income (AOR = 2.5, 95%CI 1.1, 5.2)and delivered at health facilities (AOR = 3.8, 95%CI 1.9, 7.3)were significantly associated with full-immunization.

### Conclusion

Coverage of full immunization was found to be low compared to the targets set in the Global Vaccine Action Plan(2011–2020).Two-third of the participants has poor knowledge about

**Funding:** The authors received no specific funding for this work.

**Competing interests:** The authors have declared that no competing interests exist.

**Abbreviations:** DPT, Diphtheria, Pertussis, Tetanus; EDHS, Ethiopia demographic and health survey; EPI, Expanded program on immunization; GVAP, Global Vaccine Action Plan; HSDP, Health sector development program.

vaccination. Urban residence, mother education, higher family income, male child and institutional delivery were factors. This study suggests that awareness creation, behaviour change on vaccination and enhancing utilization of maternal health service including delivery service, should be stressed.

## Introduction

Childhood immunization is the most effective and efficient intervention area of public health. It is currently high on both national and international policy and aid agendas [1–3]. To ensure maximum protection of children, the World Health Organization(WHO) launched the Expanded Program on Immunization(EPI) in 1974. In developing countries, the annual death of children has fallen below 10 million. Of this, immunization averted an estimated 2.5 million deaths [1,4,5]. Nevertheless, vaccine-preventable diseases (VPDs) are by far responsible for about 29% of under-five deaths each year globally. In 2018, more than thirty million children under five suffer from VPDs every year in Africa [3–6].

In 2015, an estimated 19.4 million infants worldwide were not reached with routine immunization; where 60% of them reside in ten developing countries [7,8]. In 2018, an estimated 86% of infants worldwide were vaccinated with three doses of the vaccine against diphtheria, tetanus, and pertussis(DTP3) up from 20% in 1980. While global coverage with the third dose of Haemophilus influenza-B and the hepatitis-B vaccine was estimated at 72% and 84%, respectively. Only 86% of children had received one dose of measles vaccine worldwide [9–11].

In Africa, significant progress has been achieved. Nevertheless, overall coverage rates remain low compared to the Global Vaccine action plan (GVAP: 2011–2020) targets. In 2018, coverage with DPT-3 and measles vaccine was 76% and 74% respectively. While 76% of the children were vaccinated for Haemophilus influenza-B (Hib) and hepatitis B. Nearly one-third of African countries did not achieve 80% infant PCV3 coverage [4,6,12].

In Ethiopian, according to the Ethiopian demographic and health survey (EDHS)-2016 and 2011, 39% and 24% of children were fully immunized, respectively. However, it remains below the goal of 66% set in the HSDP-IV [13–16]. In EDHS-2016, 73% and 53% of children received the first and third DPT-HepB-Hib(pentavalent) dose, respectively. More than eight children of every ten (82%) received the first dose of polio, but only about four in ten (44%) received the third dose [12,14,15].

In the Somali regional state, full-immunization coverage increased from 2.8% in EDHS-2005 to 22% in EDHS-2016. However, these numbers were very low compared to the other regions [14–17]. Nevertheless, studies on factors for low full immunization and reasons for discontinuation are scarce in the region. Hence, this study was intended to assess the coverage and factors affecting the full immunization status among children 12–23 months of age in the Somali region and enable to generate data that could be used for better planning and strengthening of immunization services.

## Methods

### Study design, setting and period

Community based cross-sectional study design was employed to assess coverage and associated factors of full immunization among children 12–23 month of age in Somali Region, from October 1–30, 2018.Somali region is pastoralist and an agro-pastoralist region in Eastern Ethiopia.

## Population, sample size determination and sampling procedure

All children aged 12–23 months with their mothers/caregivers were the source population. Study populations were children aged 12–23 months with their mothers/caregivers residing in randomly selected Kebele's of Somali region.

**Inclusion criteria.** Mothers with at least one child aged between 12–23 months who did take at least one dose of any vaccine were included.

**Exclusion criteria.** Those mothers who were unable to respond or very sick were excluded.

The sample size was determined by using single population proportion formula by considering the assumptions of 95% CI, 5% margin of error, design effect of 1.5, non-response rate (10%)and national coverage of full immunization (39%) [15] giving a final sample of 612. A cluster random sampling method was employed and the number of clusters was decided before data collection. Deghabur district was selected by lottery method. The lists of thirty Kebele's were taken from the administrative bodies of district and town. Then, elevenurban and rural Kebele's were selected by simple random sampling (a lottery method) and a total of 1876 households were found in these Kebele's. Each Kebele's were considered as one cluster and 60 households were selected from each of five rural Kebele's and the rest of the households were selected from the six urban Kebele's. The lists and number of households for each Kebele's was found for all selected Kebele's. in each Kebele the first household was selected randomly. The subsequent households were selected by systematic random sampling. For those households with more than one eligible child, one child was taken by lottery method.

## Data collection

A structured questionnaire was developed from DHS and other literatures in English and was translated to the local language (Somali language) (S1 File) [15,17]. The questionnaire includes; information on socio-demographic and economic status, child characteristics, reproductive/obstetric history, accessibility of vaccination service (travel time), immunization histories of children, maternal knowledge on immunization, and reasons for defaulting. Thirteen closed ended questions were developed to assess maternal knowledge towards child immunization. The content validity of the questionnaire was achieved by reviewing the previous similar studies. Pretest was carried out on 5% of respondents of the total sample in Kebridahir town. The data was collected based on the availability of immunization card and mothers/caretakers verbal report. In the selected households, mothers/caretakers of the child were asked for the presence of child's immunization card. For the child with immunization card, the information on the doses and types vaccine received by the child was copied from the card. If immunization card was unavailable for the child, the mothers/caretakers were asked for immunization history.

## Measurement

**Immunization status:** being fully vaccinated or not fully-immunized.

**Full immunization:** a child who received all basic vaccinations: One dose of BCG vaccine, three doses of Pentavalent, three doses of Polio vaccine, two doses Rota vaccine, three doses of Pneumococcus vaccine, one dose of Measles vaccine [18].

**Coverage by card only:** Coverage calculated with numerator and denominator based only on documented dose, excluding from the numerator those vaccinated by history.

**Coverage by card plus history:** Coverage calculated with numerator based on card and mother's report.

**Full Immunization coverage:** Proportion of children took all the recommended basic vaccination.

**Kebele:** is the smallest administrative unit in Ethiopia.

**Dropout rate (DoR):** is the rate difference between the initial vaccines (BCG or pentavalent one) and the final vaccines (Pentavalent three or Measles).

**Good Knowledge:** If a mother scored above the mean score for those questions related to vaccination and vaccine preventable diseases, considered to be good knowledgeable.

**Poor Knowledge:** If a mother scored below the mean score for those questions related to vaccination and vaccine preventable diseases, considered to be good knowledgeable.

## Data processing and analysis

The data was cleaned, edited and entered into Epi data version 4.1. Then, the data was exported to SPSS window version 20 for analysis. Descriptive statistics was done by computing proportions and summary statistics. Chi-square testing were used and normality were checked. Binary logistic regression model was employed to identify associated factors. Initially, bivariate logistic regression analysis was done and crude odd ratio (COR) with 95% CI was computed. In the Bivariate analysis, variables with a p-value of below 0.2 were included in the multi-variable logistic regression analysis. Adjusted odd ratios with 95% CI were calculated and factors with a p-value less than 0.05 were declared as independent predictors. Model goodness of fit was checked by Hosmer-Leme show goodness-of-fit test.

## Ethical approval and consent to participate

Written ethical approval letter was taken from Jigjiga University Research Ethics Review Committee (S1 Fig). Permission letter was also sought from Somali Region Health Office. Written consent (S1 File) was asked from each study participants (mothers/caregivers of children's aged 12–23 months before data collection. They were informed about the objective of the study, confidentiality of their data and the right to refuse participation (S1 File). Mothers with incompletely vaccinated child were counseled to complete the immunization as per the schedule.

## Results

### Socio-demographic characteristics of the mothers/caregivers

Six hundred two mothers of children aged 12–23 months of age were successfully interviewed, yielding a response rate of 98.4%. Nearly half 298 (49.5%) of participants were rural dwellers. More than half 312 (51.8%) of the respondents were in the range of 20 to 29 years with the median of 28.5 (SD ±5.2). The majority of respondents 517 (85.9%), 537 (89.2%) and 485 (80.6%) are Somalis in Ethnicity, Muslims in religion and married in marital status, respectively.

Only 74 (12.3%) of the respondents achieved secondary education and above 12.Two third 399 (66.3%) of the respondents were housewives. Regarding the average monthly family income, nearly three fourth of 447 (74.9%) of the households get 5000 Ethiopian birr and below.

### Reproductive history and child characteristics

Two hundred ninety two (48.5%) of children in the study are males. More than half of children were found between the range of 12–15 months of age with median age of 15. The average family size was 6.6 per household. Regarding antenatal care, only 205 (34.1%) of respondents had four antenatal care visits. One third of the respondents gave their last birth in the health institutions (Table 1).

**Table 1. Reproductive history and child characteristics of mothers/caregivers of children aged 12–23 months in Somali region, Eastern Ethiopia, 2018.**

| Variables | | Frequency(n = 602) | Percentage |
|---|---|---|---|
| Child sex | Male | 292 | 48.5 |
| | Female | 310 | 51.5 |
| Child alive | ≤ 3 | 192 | 31.9 |
| | 4–6 | 295 | 49.0 |
| | >6 | 115 | 19.1 |
| Family size | 1–3 | 63 | 10.5 |
| | 4–6 | 140 | 23.3 |
| | ≥ 7 | 399 | 66.3 |
| Child age | 12–15 | 349 | 58.0 |
| | 16–19 | 163 | 27.1 |
| | 20–23 | 90 | 15.0 |
| ANC | Yes | 464 | 77.1 |
| | No | 138 | 22.9 |
| Number of ANC visits (n = 464) | 1–2 | 109 | 23.5 |
| | 3–4 | 355 | 76.5 |
| TT Vaccine | Yes | 477 | 79.2 |
| | No | 125 | 20.8 |
| Number of TT vaccine (n = 477) | 1–2 | 277 | 58.0 |
| | 3–5 | 200 | 42.0 |
| Place of delivery | Home | 252 | 41.9 |
| | Health institution | 350 | 58.1 |
| Birth Order | First | 63 | 10.5 |
| | Second | 27 | 4.5 |
| | Third | 96 | 15.9 |
| | Fourth and above | 416 | 69.1 |

### Vaccination service availability and access

More than two third 424 (70.4%) of the respondents reported that there is a nearby heath facility which render vaccination service. For more than half 234(55.2%) and nearly half 201 (47.4%) of respondents an average travel time to reach the nearby health facility and an average waiting time was 15 to 30 minutes, respectively (Table 2).

### Knowledge on vaccination/Vaccine preventable diseases

The majority 528 (87.7%) of study participants ever heard about vaccination and, health personnel were the most frequent source of information for 349 (66.1%) of the respondents. Only 214 (35.5%) of the respondents know correct number of sessions to complete immunization; while 318 (52.8%) and 277 (46.0%) of mothers/caregivers Know correct age to begin immunization and correct age to complete immunization, respectively.

### Immunization status of children

Of all mothers/caregivers of children who ever took one or more dose of vaccine, nearly half 293 (48.7%) of them retained vaccination card. Of all children who were involved in the study, 256(42.5%) of them have completed all of the recommended vaccination by history (maternal recall) and plus Card. While from the total of two hundred thirty nine mother who retained vaccination cards, 87 (29.7%) of them completed the recommended vaccination.

**Table 2. Vaccination service availability and accessibility in Somali region, Eastern Ethiopia, 2018.**

| Variables | | Frequency(n = 602) | Percentage |
|---|---|---|---|
| **Presence of nearby health facility for vaccination service** | Yes | 424 | 70.4 |
| | No | 178 | 29.6 |
| **Type of Health facility (n = 424)** | Health center | 104 | 24.5 |
| | Hospital | 143 | 33.7 |
| | Health post | 169 | 39.9 |
| | Private clinic | 8 | 1.9 |
| **Travel time to the nearby health facility (n = 424)** | <15min | 71 | 16.7 |
| | 15-30min | 234 | 55.2 |
| | 31 to 60 min | 81 | 19.1 |
| | above 60min | 38 | 9.0 |
| **Waiting time (n = 424)** | <15min | 166 | 39.2 |
| | 15-30min | 201 | 47.4 |
| | 31-60min | 57 | 13.4 |
| **Functional refrigerator (n = 424)** | Yes | 244 | 57.5 |
| | No | 180 | 42.5 |
| **Defaulter tracing (424)** | Yes | 74 | 17.4 |
| | No | 350 | 82.6 |

**Immunization coverage by card only.** Of all respondents who retained vaccination card, 194 (66.2%) of children took BCG vaccine. Two hundred thirty-six (80.5%) of children took OPV1, while only 125 (42.7%) took OPV3. More than three fourth (79.9%) of respondents took pentavalent one and Rota vaccine one. While two hundred thirty-two (79.2%) of respondents took PCV one, only 118 (40.8%) of them took the third dose. Eighty-seven (29.7%) of them children were completed the recommended Vaccination (Fig 1).

**Immunization coverage by history/recall plus card.** According to vaccination card and maternal recall, from the total of 602 children, 249 (41.4%) of them were completed their immunization (fully vaccinated). The majority (77.7%) of children were vaccinated for BCG. More than nine out of ten children took OPV1, while only 269 (44.7%) took OPV3, where 50.9% dropout rate from OPV1 to OPV3. Similar trends were found with pentavalent and PCV; decrement was seen from the first dose to the last dose. Five hundred thirty-nine (89.5%) took pentavalent one, while 267 (44.3%) of the respondents took Pentavalent 3, with 50.4%

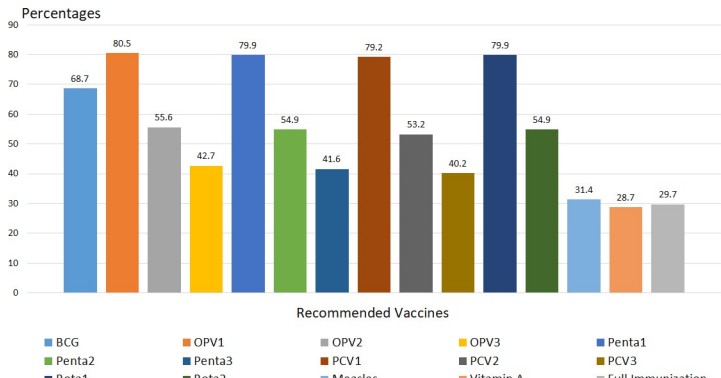

**Fig 1. Full immunization coverage by card only, among children 12–23 month of age, in Somali region, Eastern Ethiopia 2018.** *(Note*: (BCG- Bacillus calmette-guerin, OPV- Oral polio vaccine, Penta-pentavalent, Rota- Rota vaccine, PCV-Pneumococcal conjugate vaccine).

dropout rate. Nearly half (46.3%) of children took measles vaccine; where 40.4% BCG to measles dropout rate were seen (Fig 2).

## Reasons for not fully vaccinated

Mothers/care givers with incompletely vaccinated child were asked for reasons for drop out. Of the total children, 353 (58.6%) of were not fully vaccinated. Not knowing to come back for next visits were the most frequently mentioned reason for 197 (55.8%) mothers/care givers, followed by vaccination site far away 112 (31.7%) (Fig 3).

## Factors associated with full immunization

In the bivariate logistic regression twelve variables were found to be candidates for multivariable logistic regression. Nevertheless, in the multiple logistic regression: residence, maternal marital status, maternal education, average family income, sex of the child, place of delivery and travel time to the nearby facility were significantly associated with full immunization.

Children of mothers/care takers who reside in the urban were two times AOR: 2.0, 95%CI [(1.0, 3.9)] more likely to fully vaccinate their child compared to children of mothers/care takers in the rural. Children's of mothers who are primary educated were two times AOR: 2.2, 95% CI [(1.0, 5.0)] more likely to be vaccinated compared to children's of mothers who were illiterate. Children's of mothers who are married were four times AOR: 4.2, 95%CI [(1.0, 18)]more likely to be fully vaccinated compared to children's of mothers who are single. Children's who are from a family with an average monthly income of 5000 birr and above were more than two times AOR: 2.5, 95% CI [(1.1, 5.2)] more likely to be vaccinated than children's of a family with a monthly of 1000 and below. Male children were nearly two times AOR: 1.7, 95%CI [(1.0, 2.7)] more likely to be fully vaccinated than their counter parts. Children's of mothers who delivered at health facilities were nearly four times AOR: 3.8, 95%CI [(1.9, 7.3)] more likely to complete their vaccination. Mothers/caregivers who traveled to the nearby health facility in less than 30 minutes were more than two times AOR: 2.6, 95%CI [(0.8, 8.3)] more likely to fully vaccinate their child than mothers/caregiver who travel for above an hour (Table 3).

## Discussion

This study determined the prevalence of full immunization and associated factors among children aged 12–23 months. The prevalence of full immunization was 29.7% by card only and

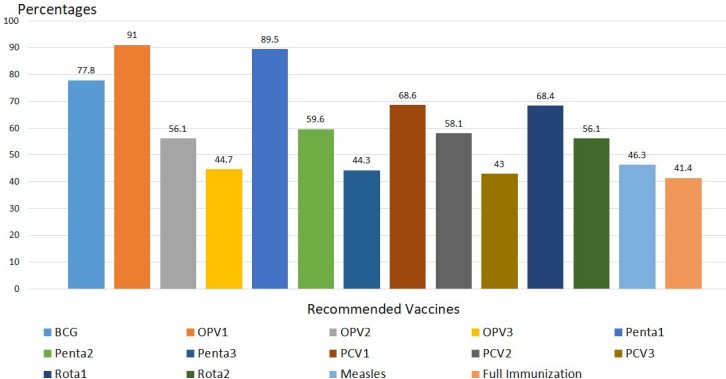

**Fig 2. Full immunization coverage by maternal recall plus vaccination card, among children 12–23 month of age, in Somali region, Eastern Ethiopia 2018.** *(Note*: (BCG- Bacillus calmette-guerin, OPV- Oral polio vaccine, Penta-pentavalent, Rota- Rota vaccine, PCV-Pneumococcal conjugate vaccine).

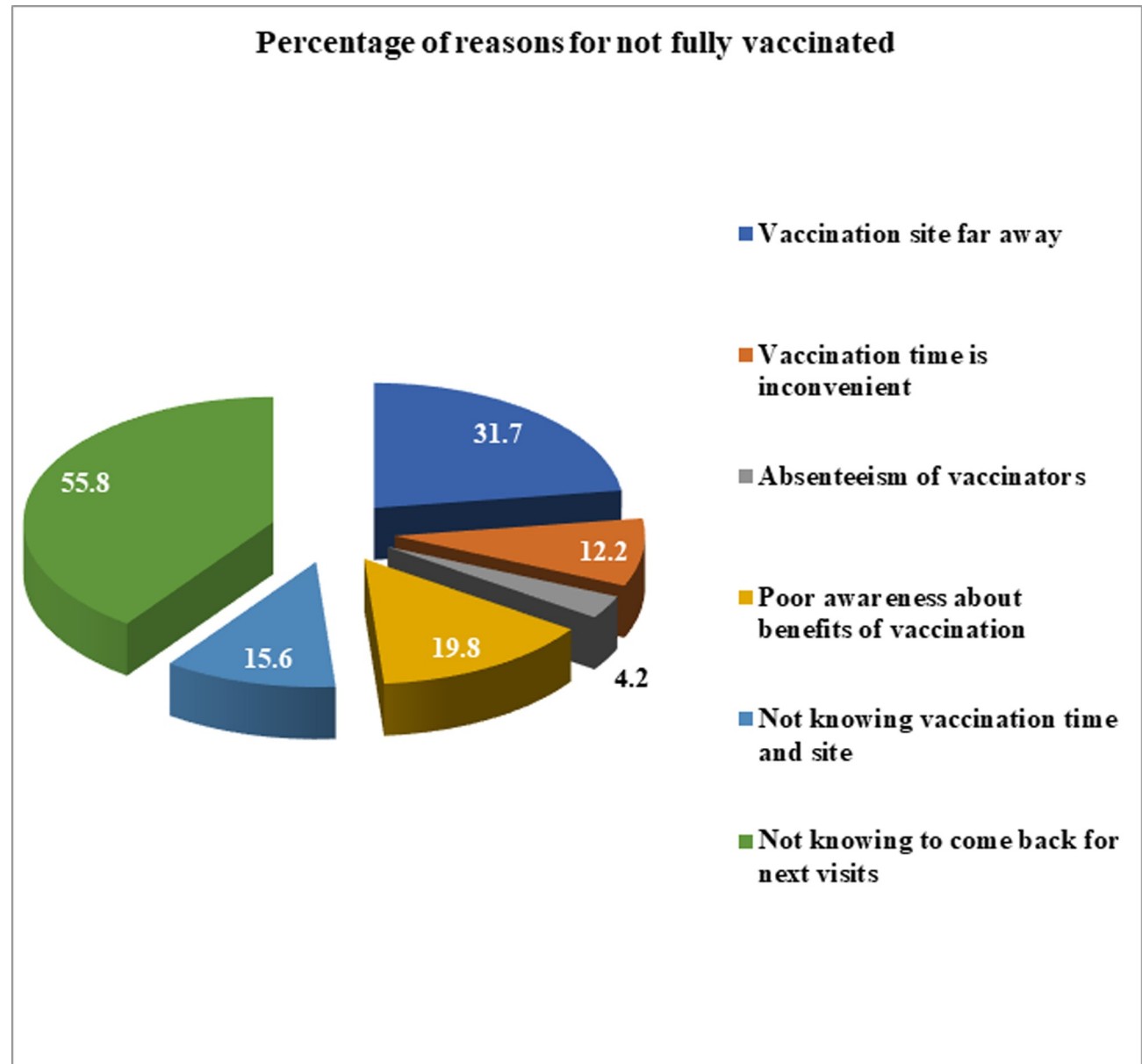

**Fig 3. Reason for not fully immunized among children 12–23 month of age, in Somali Region Eastern Ethiopia 2018.**

41.4% by history/recall plus card. It was higher compared to EDHS-2016 Somali region prevalence (22%), and Jijiga district (36%). This difference is expected as progress has been made since EPI initiated and intensive vaccination campaigns before the study [15,19]. Nevertheless, it was low compared to the goal of 66% coverage set in the HSDP-IV and other district level studies in Ethiopia [11,20,21]. It was also low compared to low-and middle-income countries reported coverage ranges from 63%-90%, and 80% targets set in the GVAP(2011–2020) and UNICEF Strategic Plan(2018–2021).The difference is expected as our study place is a pastoralist and agro-pastoralistarea. Improvements in program performance in the aforementioned countries were reported [3,22–24].

**Table 3. Determinants of full immunization among children 12–23 months of age, in Eastern Ethiopia 2018.**

| Variable | | Full immunization (n = 602) | | COD (95% CI) | AOR 95% CI |
|---|---|---|---|---|---|
| | | Yes | No | | |
| **Residence** | Urban | 180 | 124 | 4.8(3.4, 6.8) | 2.0 (1.0, 3.9) * |
| | Rural | 69 | 229 | 1 | |
| **Maternal marital status** | Single | 4 | 17 | 1 | |
| | Divorced | 14 | 20 | 2.9(0.8, 10.7) | 8.9(1.5, 54.2) * |
| | Married | 217 | 268 | 3.4(1.1, 1.3) | 4.2(1.0, 18) * |
| | Widowed | 14 | 48 | 1.2(0.3, 4.3) | 11.3(1.5, 83.2) |
| **Educational status** | Illiterate | 133 | 266 | 1 | |
| | Read and write | 39 | 40 | 1.9(1.1, 3.1) | 1.1 (0.5, 2.2) |
| | Primary | 32 | 18 | 3.5(1.9, 6.5) | 2.2 (1.0, 5.0) * |
| | Secondary | 27 | 21 | 2.5(1.4, 4.7) | 1.2 (0.4, 3.6) |
| | Above 12 | 18 | 8 | 4.5(1.9, 10.6) | 1.3 (0.3, 5.1) |
| **Average monthly family income** | ≤1000 | 22 | 85 | 1 | |
| | 1001–2500 | 58 | 83 | 2.7(1.5, 4.8) | 1.5(0.6, 3.5) |
| | 2501–4999 | 73 | 112 | 2.5(1.4, 4.4) | 2.0(0.9, 4.4) * |
| | ≥5000 | 96 | 73 | 5(2.9, 8.9) | 2.5(1.1, 5.2) * |
| **Sex of the child** | Male | 141 | 151 | 1.7(1.2, 2.4) | 1.7(1.0, 2.7) * |
| | Female | 108 | 202 | 1 | |
| **Place of delivery** | Home | 55 | 197 | 1 | |
| | Health institution | 194 | 156 | 4.4(3.1, 6.4) | 3.8(1.9, 7.3) * |
| **Knowledge about Vaccination and VPD** | Good knowledge | 141 | 97 | 3.4(2.4, 4.8) | 1.5(0.9, 2.6) |
| | Poor knowledge | 108 | 256 | 1 | |
| **Travel time to the nearby health facility(n = 424)** | <15min | 39 | 32 | 5.4(2.1, 13.8) | 2.9(0.8, 9.4)* |
| | 15-30min | 138 | 96 | 6.3(2.7, 15.0) | 2.6(0.8, 8.3) |
| | 31–60 min | 25 | 56 | 1.9(0.7, 5.0) | 0.9(0.3, 3.3) |
| | Above 1hr | 7 | 31 | 1 | |

* significant (P-value <0.05).

This study found that a dropout rate ranging from 40% to 50.9%. This rate was higher than the reported National dropout rate (20% to 25%), and other findings in Ethiopia [15,16,25]. This was possibly due to the mothers/care-takers lack of access for maternal health service.

Children of mothers who reside in urban were found to be a positive predictor of full immunization. It is supported by evidences from Ethiopia, Nigeria, Pakistan and Myanmar [15,26–28]. Children's of mothers who weremarried were four times more likely to be fully vaccinated. This is supported by others evidences [26,29]. This is may be due to married women get husband support in decision making. This study also showed higher full immunization coverage for those who delivered at health institution compared to home delivery. It is comparable with findings in Ethiopia, Kenya, Philippines and Pakistan. This shows increased contact with the healthcare facility would improve full immunization [21–23,30]. Male children were more likely to be fully immunized than females. This could be due topriority is given to males, in most low and middle-income countries including Ethiopia [15,20,31].

This study and others have found that household economic status was predictor of full-immunization [15,20,32]. This can be justified that children born to economically better households have more chance of being fully vaccinated.

## Conclusions

Both Fullimmunization by card only (29.7%) and by maternal recall plus card (41.4%) is low compared to the targets set in the Global Vaccine Action Plan, 2011–2020, the UNICEF Strategic Plan, 2018–2021 and other district level studies [9]. Only 39.5% of the study participants have good knowledge about immunization session correct age to start and finish vaccination. Urban residence, marital status of the mother, primary educational status of the mother, average family income, sex of the child, below 30 minute travel time to nearby facility and institutional delivery were the significantly associated with full immunization status of children age 12–23 months. On the other hand, not knowing to come back for next visits (55.8%), vaccination site far away (31.7%) and poor awareness about benefit of vaccination (19.8%) were frequently mentioned reasons for incomplete vaccination. This study suggests governmental and non-governmental organizations working on immunization locally and nationally, need to promote knowledge on proper immunization session, correct age to start and finish vaccination. National and regional programs should strive to increase accesses and utilization of maternal health service like delivery service, which have direct impact on child full immunization.

## Limitations

Assessing vaccination status based on maternal recall is liable for recall bias. Since cross-sectional study design was employed, it doesn't show temporal relationships between factors. The data was collected by interviewers that can potentially introduces social desirability bias.

## Supporting information

**S1 Fig. Ethical approval letter.**
(TIF)

**S1 File. Annex.**
(PDF)

**S2 File. Data set in SPSS version-23.**
(SAV)

## Acknowledgments

We would like to thank Jigjiga University and Somali region health office for their close support. We are grateful for study participants, supervisors and data collectors for their willingness and cooperation during data collection and field work.

## Author Contributions

**Conceptualization:** Zemenu Shiferaw Yadita.

**Data curation:** Zemenu Shiferaw Yadita.

**Formal analysis:** Zemenu Shiferaw Yadita, Liyew Mekonen Ayehubizu.

**Investigation:** Zemenu Shiferaw Yadita, Liyew Mekonen Ayehubizu.

**Methodology:** Zemenu Shiferaw Yadita, Liyew Mekonen Ayehubizu.

**Project administration:** Zemenu Shiferaw Yadita.

**Software:** Zemenu Shiferaw Yadita.

**Validation:** Zemenu Shiferaw Yadita.

**Writing – original draft:** Zemenu Shiferaw Yadita.

**Writing – review & editing:** Liyew Mekonen Ayehubizu.

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
