## [Decision Letter · Decision Letter 0]

20 May 2021

PONE-D-21-01555

Full immunization coverage and associated factors among children aged 12-23 months in Somali Region, Eastern Ethiopia

PLOS ONE

Dear Dr. Yadita,

Thank you for submitting your manuscript to PLOS ONE. After careful consideration, we feel that it has merit but does not fully meet PLOS ONE’s publication criteria as it currently stands. Therefore, we invite you to submit a revised version of the manuscript that addresses the points raised during the review process.

I encourage you to carefully review and address the concerns and comments raised by the reviewers. Among other issues raised, in this revision, please pay particular attention to the methodological and language concerns raised by Reviewer #2. These revisions are intended to improve the quality of the article.  

We look forward to receiving your revised manuscript.

Kind regards,

Comfort Z Olorunsaiye, Ph.D

Academic Editor

PLOS ONE

Journal Requirements:

3. In the Methods, please clarify that participants provided oral consent. Please also state in the Methods:

- Why written consent could not be obtained

- Whether the Institutional Review Board (IRB) approved use of oral consent

- How oral consent was documented

- Whether consent was obtained from the guardians/parents of minors

For more information, please see our guidelines for human subjects research: https://journals.plos.org/plosone/s/submission-guidelines#loc-human-subjects-research

4.We note that you have indicated that data from this study are available upon request. PLOS only allows data to be available upon request if there are legal or ethical restrictions on sharing data publicly. For information on unacceptable data access restrictions, please see http://journals.plos.org/plosone/s/data-availability#loc-unacceptable-data-access-restrictions.

Reviewers' comments:

Reviewer's Responses to Questions

**Comments to the Author**

1. Is the manuscript technically sound, and do the data support the conclusions?

Reviewer #1: Yes

Reviewer #2: Partly

2. Has the statistical analysis been performed appropriately and rigorously? 

Reviewer #1: Yes

Reviewer #2: No

3. Have the authors made all data underlying the findings in their manuscript fully available?

Reviewer #1: Yes

Reviewer #2: No

4. Is the manuscript presented in an intelligible fashion and written in standard English?

Reviewer #1: Yes

Reviewer #2: No

5. Review Comments to the Author

Reviewer #1: This is a great paper, I have enjoyed reading it.

Authors are encouraged to read through the manuscript for any typographic errors. More specifically on page 15; line 203 where figure 1 is indicated twice.

Reviewer #2: Topic: Full immunization coverage and associated factors among children aged 12-23 months in Somali Region, Eastern Ethiopia.

Version 1

General comments

Abstract:

1. Better to adhere the guide line of PLoS one journal. All major sections is needed

2. The knowledge gap is not well addressed that needs to be explained more at abstract section

3. Word consistency “associated factors or predictor factors”

4. At method section, say something about measurement and type of model for different outcome interest of the study.

5. At result section, result should be stick to the objective of the study

Introduction

1. Authors should briefly explain what has been done so far. To be more interesting, author need to consider “what is the additional knowledge has is this study going to generate”? The knowledge gap is not well addressed in the introduction of the manuscript and needs to be explained more.

2. Explain why more work/research is necessary. What contribution to knowledge that the research will make and its place in current debate or technological advances

3. Some references are olds, needs to be justify the research question or problem by using current evidences

4. There are some studies conducted in the study area, what additional knowledge gap is addressed by this study.

5. A grammatical and linguistic edit is essential for this manuscript, as the numerous issues are apparent in it and make understanding the provided draft difficult

6. What is the objective of the study? Word consistency, at introduction section assess prevalence and factors, at abstract section “Identifying predictor factors, assessed full-immunization and associated factors, at method part “assess coverage and associated factors of full immunization”----these all makes confusion for the reader.

Method and materials

1. A Method part looks like master thesis protocol. Concepts should be explained based on the guideline of the journal

2. Authors said “study populations were randomly selected children aged 12-23 months with their mothers/caregivers residing in the Somali region”. How randomly select the study population? What random method means?

3. What are the inclusion and exclusion criteria

4. At sampling technique and procedure section important information is missed, like total number of kebeles in the district, total number of the households in the selected kebeles, Authors apply systematic random sampling method, how to apply important step/information/ is not documented.

5. If authors conduct pretest, what was the modified things, you should be report and document in the manuscript

6. Your tool is adopted from DHS and another literature with some modification, so it needs tool validation, How to check reliability and validity of the tool? Cite the source of tools? The procedure of data collection?

7. What are unique variables/factors/ examined compared to the pervious available studies. All identified variables are already addressed previously studies. therefore, include variable should not be redundant

8. In the measurement section, what is the reference for the standardized tool for full immunization measurement? Also, please add whether the tool is standardized for the Ethiopian population or not. How many knowledge questions are asked? How the questions are designed?

9. Statistical analysis, replace by data processing and analysis important information are lacking like Checking of assumption (Normality and interaction effect)

10. Who approve the study? How get consent? Ethical consideration is the major concern today

Results

1. Results did not provide new knowledge in this field

2. In the regression table, some of the results indicates wide confidence interval, are you trusting these findings?

Discussions:

1. It is quite poor and repeats a lot of known facts without making any point as to how this current study contributes. A lot of results are repeated in the discussion. What are the innovative ideas, for scale up and ensure quality and safe services? Formulate clear what is innovating idea in the study.

2. Your results may be affected by social desirability bias because questionnaires were collected by interviewers; the participants were unable to remain anonymous. This should be mentioned in the limitations section.

Conclusion section:

1. Conclusion is repeated, there is significant disconnect between the results presented and the conclusions made. There was no evidence in the results or anywhere else that they looked at the possible barriers and strategies for that country under question. They can suggest but not make a hard conclusion that those strategies would work or hinder.

Recommendation: This paper is below the scope of the journal, so consider after major changes

6. PLOS authors have the option to publish the peer review history of their article (what does this mean?). If published, this will include your full peer review and any attached files.

Reviewer #1: No

Reviewer #2: No

---

## [Author Response · Author response to Decision Letter 0]

8 Aug 2021

Response to the editor and reviewers 

I. Response to the Academic Editor:

1. Maximum effort has been made to make sure that our manuscript meets PLOS ONE's style requirements.

2. Information’s regarding the questionnaire is provided in the method section. In addition, the questionnaire is submitted as a supporting information in English and Somali language.

3. Both, oral consent and written consent was taken from the mothers/caregivers of children aged 12-23 months. This information’s are included in the method section. 

4. All data supporting the findings is submitted with the manuscript. The data set for this article is openly accessible without restriction. The data set is submitted as supporting information file as SPSS version 23 data set.

II. Response to the reviewer #1

1. Maximum efforts have been made to make sure the data supports the conclusion.

2. Both descriptive and analytical statistical analysis were performed thoroughly. 

3. All data supporting the findings is submitted with the manuscript. The data set for this article is openly accessible without restriction. 

4. The manuscript is written in standard English. Typographic errors, and grammatical issues have been corrected. 

III. Response to the reviewer #2

I. Abstract:

1. The abstract is developed in line with PLoS one journal guide line. 

2. The knowledge gap, the statistical models and measurements are clearly presented in the abstract. 

II. Introduction:

1. It clearly shows the knowledge gap. In Ethiopia, progresses on child immunization has been made; however, the full immunization coverage is below the national and global targets. On the other hand, evidence on full immunization and associated factors are very scarce in the hard to reach regions of Ethiopia, particularly in Somali region. 

2. The introduction clearly presented assessment of full immunization and associated factors would be a new knowledge for the study area. Hence, this study will serve as an important evidence on child full immunization in the Somali region. 

3. Some of the old references were used to define concepts; but not to justify the problem gap.

4. There are some studies on child immunization in the study area. But, they didn’t address the full immunization coverage and associated factors.

5. Grammatical errors are corrected

6. The objective of the study was presented in different ways but with similar concept

III. Method and materials:

1. The concepts in the method section were explained based on the guideline of the journal.

2. Probability sampling methods or random sampling methods were consistently used to select study participant. 

3. Inclusion and exclusion criteria for study participants are included in method section.

4. Accepted and corrected 

5. Pretest was done on the questionnaire but major modification were not needed.

6. The content validity of the questionnaire was achieved by reviewing the previous similar studies. Data collection procedure were clearly and precisely presented in the data collection section. 

7. Most variable were not addressed in the study area.

8. Accepted and corrected. 

9. Data processing and analysis issues were clearly and precisely presented.

10. The study was approved by Jigjiga University Ethical Review Board. Written consent was sought from study participants before data collection. Participants were verbally informed about the objective of the study, confidentiality of their data and the right to refuse participation. 

IV. Results 

1. Every finding in there result section is new, because it is very rare to find evidence on full immunization coverage and associated factors, in the pastoralist and Agro-pastoralist regions of Ethiopia and Africa. 

2. Some of the results in the regression table showed wider confidence intervals because of smaller cell values. However, since assumptions were checked for binary logistic regression analysis, still the findings are trusted.

V. Discussion:

1. The discussion clearly presented the realities in the one of pastoralist region of Ethiopia compared to national and global literatures. This study will be an input for local and regional quality improvement programs. 

2. Accepted and corrected.

VI. Conclusion:

1. All conclusion has been made based on the findings in the result.

2. Recommendations were suggested based on the conclusions.

---

## [Editor Report · Decision Letter 1]

8 Nov 2021

Full immunization coverage and associated factors among children aged 12-23 months in Somali Region, Eastern Ethiopia

PONE-D-21-01555R1

Dear Dr. Yadita,

We’re pleased to inform you that your manuscript has been judged scientifically suitable for publication and will be formally accepted for publication once it meets all outstanding technical requirements. 

Kind regards,

Comfort Z Olorunsaiye, Ph.D

Academic Editor

PLOS ONE

Additional Editor Comments (optional):

This manuscript will still require careful editing to address typographical and grammatical errors prior to publication. 
---

## [Editor Report · Acceptance letter]

17 Nov 2021

PONE-D-21-01555R1 

Full immunization coverage and associated factors among children aged 12-23 months in Somali Region, Eastern Ethiopia 

Dear Dr. Yadita:

I'm pleased to inform you that your manuscript has been deemed suitable for publication in PLOS ONE. Congratulations! Your manuscript is now with our production department. 

Kind regards, 

on behalf of

Dr. Comfort Z Olorunsaiye 

Academic Editor

PLOS ONE